# Validity of the French version of the Autonomy Preference Index and its adaptation for patients with advanced cancer

Isabelle Colombet[1,2], Laurent Rigal [3,4], Miren Urtizberea[1], Pascale Vinant[2], Alexandra Rouquette[3,5]*

1 Université Paris Descartes, Sorbonne Paris Cité, Paris, France, 2 AP-HP, Palliative Medicine, Cochin Hospital, Paris, France, 3 Université Paris-Saclay, UVSQ, Inserm, CESP, Paris, France, 4 Université Paris-Saclay, General Practice Department, Le Kremlin Bicêtre, France, 5 AP-HP, Bicêtre Hospital, Public Health and Epidemiology Department, Le Kremlin-Bicêtre, France

* alex.rouquette@gmail.com

**Data Availability Statement:** All relevant data are within the manuscript and its Supporting Information files.

**Funding:** This work (IC) was supported by the Fonds pour les Soins Palliatifs [Convention with

## Abstract

### Background

While patient-centered care is recommended as a key dimension for quality improvement, in case of serious illness, patients may have different expectations regarding information and participation in medical decision-making. In oncology, anticipation of disease worsening remains difficult, especially when patient's preferences towards prognosis medical information are unclear. Valid tools to explore patients' preferences could help targeting end-of-life discussions, which have been shown to decrease aggressiveness of end-of-life care. Our aim was to establish the validity and reliability of the French version of the Autonomy Preference Index (API) among patients with incurable cancer and in primary care setting. Three supplementary items were specifically developed to evaluate preparedness to anticipate disease deterioration among patients with incurable cancer.

### Methods

The psychometric properties of the API translated into French were assessed among patients consecutively recruited from January to March 2017 in the waiting rooms of 19 general practitioners (N = 391) and in an oncology (N = 187) clinic in Paris. Relationships between the newly-developed items and the API subscale scores were studied.

### Results

A three correlated factors confirmatory model (two factors related to decision-making and a factor related to information-seeking preferences) showed an acceptable fit on the whole sample and no measurement invariance issue was found across settings, age, sex and educational level. Internal consistency and test-retest reliability were acceptable for the information-seeking and decision-making subscales. One of the newly-developed items on patients' ability to anticipate a decision on the use of artificial respiration if a sudden deterioration of their illness occurred was not related to the API subscale scores.

Paris Descartes University n° J17P99CONV11]. The funders had no role in study design, data collection and analysis, decision to publish, or preparation of the manuscript.

**Competing interests:** The authors have declared that no competing interests exist.

## Conclusion

The French version of the API was found valid and reliable for use in general practice and oncology settings. The additional items on patient preparedness to anticipate disease deterioration can be of interest to ensure that patient values guide all end-of-life clinical decisions.

## 1. Introduction

Shared decision-making is a process in which a choice is jointly made by a provider and a patient or a proxy decision-maker [1]. Taking its roots in the "patient-centered care" movement in healthcare, this process was pointed to, at the turn of the millennium, as a key aim to ensure that 21st-century Health Care Systems "cross the Quality chasm" [2,3]. Consideration of patient preferences as to their level of involvement in the decision-making process has now become an ethical imperative, and has been integrated into healthcare programs and legal texts in many countries [3,4].

While patient participation in decision-making processes is essential in all medical contexts, it is particularly complex in situations of incurable illness. Many informed decisions need to be made, for example, treatment limitation or cessation, Do-Not-Resuscitate orders, place of care, etc. [5]. Information-sharing between the patient and the physician is recognized as one of the main characteristics in the definition of shared decision-making in healthcare [6]. Information on disease evolution and prognosis is a prerequisite for patients to assess the risk-benefit ratios of their therapeutic options [7].

In oncology, numerous studies have shown that patients do not receive exhaustive information on their situation maybe because delivery and receipt of this information are tricky for both parties in the sharing process [8–10]. Physicians may worry about increasing patient anxiety, as it has been shown that patients have variable expectations towards prognostic medical information [11]. This may be the reason why anticipation of disease deterioration still remains difficult for both patient and oncologist, although end-of-life discussions were already suggested several years ago to reduce the aggressiveness or invasiveness of end-of-life care by facilitating shared decisions and the traceability of do-not-resuscitate orders [12].

In this context, physicians need to adapt their communication according to patients' expectations regarding information and their desire to be involved in decisions, and also according to their preparedness to anticipate disease deterioration [13]. For that matter, the need for an assessment of these patient's preferences has been highlighted in the literature [4,13,14]. To our knowledge, three measurement tools aiming to assess both information preferences and the desire to participate in decision-making have already been used among patients with incurable or terminal cancer: 1) visual analog scales initially developed for patients in emergency wards [15,16], 2) the Krantz Health Opinion Survey, a self-administered 16-item questionnaire, initially developed for students, and concerning medical care in general with a focus on self-medication [17,18], and 3) the Autonomy Preference Index (API), a self-administered 23-item questionnaire, initially developed for patients in primary care settings [19].

The API has various advantages to be used among patients with incurable cancer over the two other measurement tools identified. First, it does not focus on self-medication contrary to the Krantz Health Opinion Survey. Second, its psychometric properties have already been studied in English and German in various populations (primary care settings, patients with asthma, mental illness, chronic pain) [20–25]. Third, its original structure allows for

adaptation depending on the context as it has already been done for psychiatric patients for example [21,24,25]. Among the 23 items of its original version, 8 items assess information-seeking (IS score) preferences and the remaining 15 items assess preferences for participation in decision-making (DM) including 6 general items used to compute the DM score and 9 items related to three clinical vignettes representing different levels of severity: the upper respiratory tract illness (URI score) is used to represent a mild condition; hypertension (HBP score) for a moderately severe condition; and myocardial infarction (MI score) for a severe life-threatening condition. In some previous studies, only the 14 items related to the IS and DM scores were used [23,25,26] while in others, vignettes were adapted to the context [20,21,24]. So in our study, we aimed to validate the API in a population of patients with incurable cancer, and to develop an additional vignette with supplementary items, specifically for these patients, to evaluate their preparedness to anticipate disease deterioration, as this was not addressed in the original API.

The working objectives of this study was thus to translate the API, to evaluate its psychometric properties (reliability and construct validity) in a population of primary care patients, as for the original version, and in a population of patients with incurable cancer. We also assessed measurement invariance which is an essential property for questionnaire, as for any measurement tool, to guarantee accurate group comparisons. According to Mokkink et al. and Milsap, "a measuring device should function in the same way across varied conditions, so long as those varied conditions are irrelevant to the attribute being measured" [27,28]. We studied measurement invariance across age, sex and education level as usually performed and recommended, across French and English languages to ensure the comparability of the scores from both language versions, and across both settings (primary care patients and patients with incurable cancer) to check the likeness of the API factor structure in these settings [28–30]. The Consensus-based Standards for the selection of health Measurement Instruments (COSMIN) guidelines were followed to report the results [31].

## 2. Methods

### 2.1. The Autonomy Preference Index and the supplementary items for preparedness to anticipate disease deterioration

A 5-point Likert scale is used to answer to the 23 items of the API (a score of 5 indicating the strongest preference) (S1 Questionnaire, S1 Table). The computation of the five scores from the API was explained in the original publication as follow: The IS and DM scores are computed as the sum of the 8 and 6 answers respectively linearly adjusted to range from 0 to 100 (strongest desire possible). The URI, HBP and MI scores are computed from the sum of the answers to the three items, linearly adjusted to range from 0 to 10 (strongest desire possible) [19].

The additional clinical vignette developed to address the preparedness of patients with advanced cancer to anticipate disease deterioration. This vignette concerns a chronic, terminal respiratory illness requiring oxygen therapy that can potentially evolve towards a sudden deterioration, requiring artificial respiration (S2 Questionnaire, Table 1). This situation was chosen to minimize the chances for a patient with advanced cancer of identifying with this situation. The three items (answers on a 5-point Likert scale) concerned the desire to participate in the advance decision on whether to use artificial respiration, preference regarding the anticipation of this decision, the ability to decide on this point at a time when the situation has not yet arisen.

**Table 1. Frequencies (%) of the answers to the items of the additional clinical vignette "preparedness to anticipate disease worsening" in the ONCO group.**

| **Additional clinical vignette**: "Suppose you are suffering from a chronic, terminal respiratory disease. At home, you need oxygen therapy all the time and your movements are limited. You know that in case of sudden deterioration (for example because of a lung infection), you may have to be put on artificial respiration (a tube connected to a machine that breathes for you, while you are asleep and unconscious), without you being able to give your opinion. Regarding the decision to use this artificial respiration:" | **N (%)** |
|---|---|
| **1—In your opinion, who should make this advance decision (at a time when the sudden aggravation has not yet occurred)?** (a single answer) | |
| · I would prefer to be left to make my own decision | 5 (3) |
| · I would rather the decision be left to me, after having taken my doctor's advice into consideration | 24 (13) |
| · I would rather decide together with my doctor | 71 (38) |
| · I would prefer to let my doctor decide, once my opinion has been taken into consideration | 56 (30) |
| · I would prefer to let my doctor decide alone | 30 (16) |
| · Missing | 1 (0) |
| **2—Is it important for you that your doctor should discuss this decision with you in advance, in anticipation of a sudden deterioration?** | |
| · Yes, absolutely | 136 (73) |
| · Mostly yes | 41 (22) |
| · Neutral | 3 (2) |
| · Mostly no | 5 (3) |
| · No, not at all | 1 (1) |
| · Missing | 1 (0) |
| **3—Do you think it is possible to express an opinion regarding this decision at a time when the situation has not yet arisen?** | |
| · Yes, absolutely | 63 (34) |
| · Mostly yes | 67 (36) |
| · Neutral | 23 (12) |
| · Mostly no | 22 (12) |
| · No, not at all | 8 (4) |
| · Missing | 4 (2) |

## 2.2. Translation process

Following the steps described in the current recommendations on the cross-cultural adaptation of questionnaires [32,33], four French experts from various disciplines (palliative care, general medicine, public health, epidemiology, biostatistics, psychometrics) with good English language proficiency and two English-French bilinguals independently translated the English version of the API into French [34]. A consensus meeting was then held to reach a consensual French version of the questionnaire, on the basis of the six independent translations. The author of the first version of the API, J. Ende, was contacted to ask for permission, but he was not available to participate in the translation process. No back-translation was performed as it is not required in this context [35]. Individual semi-structured cognitive debriefing sessions (acceptability, comprehensibility and consistent interpretation across participants) were organized with 13 subjects (7 with incurable cancer and 6 without any declared illness; 8 males; 4 under 30 years, 7 aged 30 to 70 years and 2 over 70 years) who tested this version (completion time: 3 to 10 min). Minor form changes were made on some items following the content analysis of these debriefing meetings, yielding the final French version of the API (S1 Questionnaire).

## 2.3. Study samples

Two samples of subjects were consecutively recruited from January 2017 to March 2017: 1) in the waiting rooms of 19 general practitioners involved in the general practice network of Paris-Sud University (France) and selected to ensure representation of the various social backgrounds in the Paris area (GP sample), 2) in the oncology outpatient clinic of Cochin Hospital in Paris (ONCO sample). Cochin hospital is a tertiary care hospital treating around 4500 new cancer patients each year, with an oncology ward and three other medical specialty wards (gastroenterology, pneumology, dermatology) that have an oncologic activity of care and use the oncology outpatient clinic for ambulatory anticancer treatment and follow-up. Explanations on the study were provided to all consecutive French-speaking patients aged 18 years or older, without cognitive or psychopathologic disorders, by an independent researcher, unknown to the patients in the two settings. Patients were included in the study if they agreed to participate and, for patients recruited in the oncology clinic, if their Eastern Cooperative Oncology Group (ECOG) performance status was 2 or below and if they had incurable cancer. There was no incentive to participate to this study. This study was approved by the ethics committee "Comité de Protection des Personnes Sud-Est VI" (n˚ID-RCB: 2016-A01960-51) and patients provided written informed consent to participate. Measurement invariance across the French and English language versions was studied for the IS and DM items using data from the only known study in which the API was used, involving 120 patients with incurable cancer in Australia [26].

## 2.4. Data collection

Using a self-administered questionnaire, the patients provided socio-demographic information including sex, age, educational level, profession and whether they were living with a partner or were single. In the ONCO sample, information on their cancer history and treatment was collected from medical files, while in the GP group, their perceived health status was collected using the following question: "Would you say that overall, your health is: excellent / very good / good / medium / poor?". The patients completed the French version of the API (and the additional vignette in the ONCO sample) and answered two questions on their global judgment concerning their information preferences (on a 4-point Likert scale) and their desire to participate in decisions (on a 5-point Likert scale) (Table 2). In the ONCO sample, patients were asked if they would agree to complete the API again at the time of their next scheduled visit (every 15–21 days).

The characteristics of the 578 patients included are described for each sample in Table 2. In the GP sample, subjects were younger (49±17 vs 64±12 years), more frequently women, with a lower level of education and less frequently professionals or managers. In the ONCO sample, cancer had been diagnosed for a median time of 20 (8–41) months and the primary tumour sites were lung, colon and/or rectum, pancreas and ovary for 47(25%), 27(14%), 23(12%) and 22(12%) of the patients respectively. In the GP sample, 297(76%) patients rated their health as "excellent, very good or good" and 93(24%) rated their health as "medium or poor".

## 2.5. Statistical analyses

Categorical data was summarized as frequencies (%) and quantitative data as means ± standard deviation or medians (first quartile–third quartile) as appropriate. For each item, we looked for ceiling and floor effects (threshold chosen *a priori* >95% of respondents choosing the highest and lowest categories respectively).

**2.5.1. Psychometric properties of the API.** The structural validity was studied using confirmatory factor analysis (CFA) with a robust estimator for categorical data, the Weighted

**Table 2. Characteristics and scores of the samples.**

| | GP sample | ONCO sample |
|---|---|---|
| **Characteristics—N (%)** | N = 391 | N = 187 |
| Age | | |
| · 40 years or younger | 142 (36) | 5 (3) |
| · 41 to 55 years | 113 (29) | 33 (18) |
| · 56 to 70 years | 93 (24) | 89 (47) |
| · Older than 70 years | 43 (11) | 60 (32) |
| Sex (Male) | 132 (34) | 86 (46) |
| Living as a couple | 259 (66) | 130 (71) |
| Social benefits | 52 (13) | 19 (12) |
| Education | | |
| ·Middle school or none | 78 (20) | 28 (15) |
| · High school | 136 (35) | 44 (24) |
| · Higher education | 176 (45) | 109 (61) |
| Profession | | |
| · Shopkeepers and tradesmen | 14 (4) | 14 (8) |
| · Professionals and managers | 102 (27) | 75 (43) |
| · Office, sales, and service employees | 111 (29) | 44 (24) |
| · Skilled or unskilled manual workers | 106 (27) | 32 (18) |
| · White-collar workers | 33 (9) | 9 (5) |
| · Never worked ? | 21 (5) | 6 (3) |
| **Autonomy Preference Index scores—mean (SD)** | | |
| · Information-seeking score | 86.8 (10.3) | 85.3 (13.3) |
| · Decision-making score | 47.6 (16.0) | 45.6 (17.5) |
| · Clinical vignette URI | 4.6 (1.8) | 4.2 (1.7) |
| · Clinical vignette HBP | 3.1 (1.7) | 2.5 (1.8) |
| · Clinical vignette MI | 3.5 (1.7) | 3.2 (1.8) |
| **Global judgement on information preferences—N (%)** | | |
| · I would prefer to be informed about everything | 232 (59) | 138 (76) |
| · I would prefer to be informed if I ask for i | 75 (19) | 29 (16) |
| · I would prefer to let my doctor decide what I need to be informed abou | 83 (21 | 13 (7 |
| · I would prefer not to be informed | 1 (0) | 2 (1) |
| **Global judgement on decision preferences—N (%)** | | |
| · I would prefer to be left to make my own decisions | 4 (1) | 6 (3) |
| · I would prefer to be left to decide after taking my doctor's advice into consideration | 60 (15) | 20 (11) |
| · I would prefer to make a decision together with my doctor | 201 (51) | 112 (62) |
| · I would prefer to let my doctor decide after having taken my opinion into consideration | 84 (22) | 36 (20) |
| · I would rather let my doctor decide alone | 42 (11) | 7 (4) |

GP: general practice, ONCO: oncologic service, URI: Upper respiratory tract illness, HBP: High blood pressure, MI: Myocardial infarction.

Least Square Means and Variances adjusted [36]. Two models were fitted, as they were both previously found in the literature to possess an acceptable fit: a three-factor model (8 IS items, 6 DM items, 9 clinical vignette items [24]), and a two-factor model (8 IS items, 15 DM and

clinical vignette items [19]). In the previous studies, the factor corresponding to the 8 IS items was not or poorly correlated (<0.3) to the factor(s) related to DM items [19,23,24]. Model fit was assessed using the Comparative Fit and Tucker Lewis Indices (CFI & TLI, good fit if >0.95, poor fit if <0.90, acceptable fit otherwise), the Root Mean Square Error Approximation (RMSEA, good fit if <0.06, poor fit if >0.1, acceptable fit otherwise) and models were compared using a nested model test [37].

Measurement invariance was tested consecutively across groups defined by the inclusion setting (GP or ONCO sample), age (categorized according to quartiles), sex, educational level and language version. A multigroup CFA and the classic three-step sequence were used to investigate configural, metric and scalar invariance [38,39]. We consecutively tested these three levels of invariance in fitting three different nested models having increasing constraints. For the sex invariance for example, the same model was hypothesized in both groups and the followed sequence of nested model tests was: 1) configural invariance: unconstrained factor loadings and item thresholds; 2) metric invariance: factor loadings constrained to be equal across sex groups and unconstrained item thresholds; 3) scalar invariance: factor loadings and item thresholds constrained to be equal across sex groups. Each level of measurement invariance was considered to be present if the fit indices difference, ΔCFI and ΔRMSEA, between nested models was –0.01 and 0.015 or below respectively [40–42].

Internal consistency was assessed using Cronbach's alpha and McDonald's omega coefficients (acceptable if ≥0.7) [43,44]. Test-retest reliability was assessed among patients in the ONCO sample who agreed to complete the API again at their next scheduled visit, using intraclass correlation coefficients (ICC, acceptable if ≥0.7) for scores on each API subscale [45]. To assess convergent validity, the association between API subscale scores and the patients' global judgment on their information preferences and desire to participate in decisions was evaluated using a one-way analysis of variance. Finally, for hypothesis testing, mean API subscale scores were compared, using a one-way analysis of variance or Student t-tests as appropriate between patients according to sex (*a priori* hypothesis: lower scores among men), age (lower scores among older patients), marital status (higher scores for singles) and educational level (higher scores for higher education levels).

**2.5.2. Relationships between items in the additional vignette and API subscale scores.** In the ONCO sample, the relationships between answers to the items in the additional clinical vignette and the API subscale scores were studied using Kruskall-Wallis or Mann-Whitney's tests as appropriate. Fisher's exact tests were also used to study associations with global judgments on information preferences and the desire to participate in decisions.

Statistical tests were two-sided and a p-value>0.05 was considered significant. Analyses were performed using Stata v.14 software for data management and basic statistics and Mplus v7.4 software for the confirmatory factor analysis (CFA), which implements full information maximum likelihood to handle missing data (lower than 2% whatever the item in the whole sample) [46,47].

## 3. Results

The characteristics of the 578 patients included are described for each sample in Table 2. In the GP sample, subjects were younger (49±17 vs 64±12 years), more frequently women, with a lower level of education and less frequently professionals or managers. In the ONCO sample, cancer had been diagnosed for a median time of 20 (8–41) months and the primary tumour sites were lung, colon and/or rectum, pancreas and ovary for 47(25%), 27(14%), 23(12%) and 22(12%) of the patients respectively. In the GP sample, 297(76%) patients rated their health as "excellent, very good or good" and 93(24%) rated their health as "medium or poor".

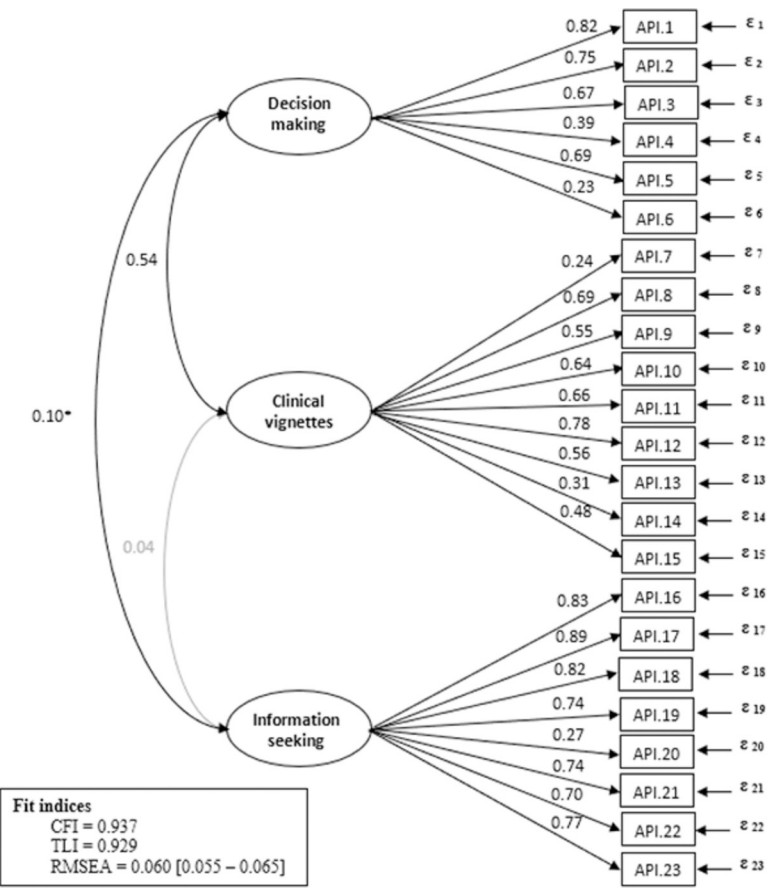

**Fig 1. Path diagrams (with standardized coefficients) for the three factor model with fit indices using a confirmatory factor analysis (robust weighted least squares [WLSMV] estimator).** Ellipses represent unobserved latent factors, rectangles represent observed variables, single-headed arrows represent the effect of one variable on another (factor loading) and double-headed arrows represent covariance between pairs of variables. Coefficients are all statistically significant with a p-value<0.001, except *p-value = 0.039 and coefficient in grey which is not statistically significant. ε: measurement error df: degree of freedom. CFI: Comparative Fit Index. TLI: Tucker Lewis Index. RMSEA: Root Mean Square Error Approximation.

Frequencies of answers to each item in the API in the two samples are summarized in S1 Table. No floor or ceiling effect was identified and there were fewer than 2.5% missing answers to each item. Scores on each of the subscales are shown in Table 2. No difference was found concerning the DM and IS scores, but significantly higher scores were observed for the URI and HBP vignettes in the GP sample than in the ONCO sample. Frequencies of answers to each of the three items in the additional vignette in the ONCO sample are shown in Table 1. A third of the sample preferred an equally shared decision with the doctor concerning the use artificial respiration in this fictional situation, and three quarters would wish to address this point with their doctor in advance, and thought that it was possible to give their opinion on this decision at a time when the situation had not yet arisen.

The three-factor CFA model shown in Fig 1, provided an acceptable fit to the data (CFI = 0.94, TLI = 0.93 and RMSEA = 0.060, 95%CI: [0.055 to 0.065]), better (p<0.001) than the fit of the two-factor model (CFI = 0.65, TLI = 0.61 and RMSEA = 0.142, 95%CI: [0.137 to 0.146]). As revealed by the ΔCFI and ΔRMSEA, no measurement invariance issue was found across groups defined by inclusion setting, age, sex, educational level, and language version, as

the highest level of measurement invariance studied (scalar invariance) was reached (S2 Table).

Cronbach's alpha coefficients were 0.69 for the 6 DM items, 0.73 for the 9 items related to the clinical vignettes and 0.71 for the 8 IS items. McDonald's omega coefficients were 0.72 for the 6 DM items, 0.73 for the 9 items related to the clinical vignettes and 0.75 for the 8 IS items. For the assessment of test-retest reliability, 96 patients from the ONCO sample completed the API again at their next scheduled visit (mean time from baseline: 17±4 days). The ICCs were 0.80 (95%CI: 0.70 to 0.87) for the DM score, 0.59 (95%CI: 0.45 to 0.71) for the URI vignette score, 0.68 (95%CI: 0.55 to 0.77) for the HBP vignette score, 0.59 (95%CI: 0.44 to 0.71) for the MI vignette score and 0.72 (95%CI: 0.70 to 0.87) for the IS score.

Results concerning convergent validity and hypothesis testing are shown in Table 3. Good convergent validity was observed for every subscale with statistically higher scores in groups defined by a stronger desire for decision-sharing and information. The a *priori* hypotheses were supported by the data for all patient characteristics studied on most of the subscales.

The relationships between answers to the items in the additional clinical vignette and the API subscale scores are shown in Table 4. The desire to participate in the advance decisions was strongly and positively related to DM and the vignette subscale scores (p<0.001), but not to the IS score (p = 0.223); preference regarding the anticipation of this decision was related to the IS score (p = 0.010) but not to other API scores (p>0.05), and the ability to decide on this point at a time when the situation had not yet arisen was not related to any of the API subscale scores (p>0.05). The same relationships (or lack of relationships) were observed with the global judgement on information preferences and desire to participate in decisions.

## 4. Discussion

In this study, the French version of the API showed adequate psychometric properties for use among patients in primary care settings or among patients with incurable cancer. The additional vignette specifically developed for use among patients with advanced cancer brought additional information on the patients' preparedness to anticipate disease deterioration: while its first item (desire to participate in the advance decision to use artificial respiration) and second item (preference regarding the anticipation of this decision) correlated with the DM and IS subscale scores in the API, its third item (addressing patients' "ability to decide on this issue at a time when the situation has not yet arisen") did not correlate with any of the API subscales. Indeed, the API do not address the question of anticipation of end-of life decisions.

Interestingly, whereas the practice of end-of-life discussions is far from being common in France and very few patients have written their living wills [5], very few patients (0 to 2%) failed to answer these items. Research on end-of-life quality of care has recently shown that Advance Care Planning (ACP) is beneficial for shared decision-making, the traceability of do-not-resuscitate orders, and the reduction of aggressive end-of-life care [12,26,48], but that it can also disrupt coping mechanisms for some patients. Indeed, results from the Coping with Cancer study suggested that patients with such psychosocial factors as emotional numbness may have their fears rather exacerbated by end-of-life discussions, resulting in unreasonable demands of care and life-maintaining treatments [49]. Educational initiatives to improve communication and enhance implication in decision-making among seriously ill patients are therefore needed and are currently being developed in protocols interestingly involving both healthcarers and patients/caregivers perspectives [50–52]. Since this module provides additional information on patient preferences for anticipation, a theme that is not addressed by the API, and since it is well accepted by patients, our clinical vignette can be used in conjunction

**Table 3. Convergent validity and hypothesis testing: mean (SD) scores to the API subscales according to patients' global judgement and characteristics (N = 578).**

| | | Scores to the API subscales | | | | |
|---|---|---|---|---|---|---|
| | N (%) | DM | URI | HBP | MI | IS |
| **Global judgement on decision preferences** | | | | | | |
| · … to be left to make my own decisions /… to decide after taking my doctor's advice into consideration. | 90 (16) | 58.8 (16.4) | 5.3 (1.8) | 3.9 (1.7) | 4.1 (1.8) | |
| · … my doctor and I decide together | 313 (55) | 48.0 (15.0) | 4.6 (1.6) | 3.0 (1.7) | 3.5 (1.6) | |
| · … to let my doctor decide after taking my opinion into consideration / … to let my doctor decide alone | 169 (30) | 38.4 (14.3) | 4.0 (1.9) | 2.2 (1.6) | 2.9 (1.7) | |
| p-value* | | <0.001 | <0.001 | <0.001 | <0.001 | |
| **Global judgement on information preferences** | | | | | | |
| · … to be informed about everything | 370 (65) | | | | | 89.2 (9.3) |
| · … to be informed if I ask for it | 104 (18) | | | | | 82.4 (11.6) |
| · … to let my doctor decide what I need to be informed about / … not to be informed about | 99 (17) | | | | | 80.1 (13.9) |
| p-value* | | | | | | <0.001 |
| **Gender** | | | | | | |
| · Male | 218 (38) | 43.7 (15.1) | 4.4 (1.8) | 2.6 (1.7) | 3.3 (1.8) | 85.3 (11.7) |
| · Female | 360 (62) | 49.0 (17.0) | 4.6 (1.7) | 3.1 (1.8) | 3.5 (1.7) | 87.0 (11.1) |
| p-value** | | <0.001 | 0.310 | 0.002 | 0.201 | 0.081 |
| **Age** | | | | | | |
| · 40 years or younger | 147 (25) | 50.4 (16.2) | 4.5 (1.6) | 3.2 (1.7) | 3.4 (1.6) | 88.1 (9.0) |
| · 41 to 55 years | 146 (25) | 48.0 (15.3) | 4.5 (1.7) | 3.0 (1.6) | 3.4 (1.7) | 86.9 (10.7) |
| · 56 to 70 years | 182 (31) | 46.4 (16.7) | 4.5 (1.9) | 2.8 (1.5) | 3.3 (1.9) | 84.8 (13.0) |
| · Older than 70 years | 103 (18) | 41.4 (16.9) | 4.3 (1.8) | 2.5 (1.8) | 3.5 (1.8) | 85.7 (12.0) |
| p-value* | | <0.001 | 0.655 | 0.039 | 0.884 | 0.066 |
| **Living as a couple** | | | | | | |
| · No | 186 (32) | 47.8 (16.3) | 4.8 (1.9) | 3.2 (1.8) | 3.5 (1.7) | 85.2 (11.2) |
| · Yes | 389 (68) | 46.5 (16.6) | 4.3 (1.7) | 2.8 (1.7) | 3.3 (1.7) | 86.9 (11.4) |
| p-value** | | 0.379 | 0.001 | 0.004 | 0.176 | 0.095 |
| **Education level** | | | | | | |
| · Middle school or none | 106 (19) | 40.8 (16.3) | 4.5 (2.1) | 2.5 (1.9) | 3.7 (1.8) | 83.9 (12.2) |
| · High school | 180 (32) | 44.3 (15.9) | 4.5 (1.6) | 2.9 (1.6) | 3.4 (1.8) | 86.5 (10.5) |
| · Higher education | 285 (50) | 50.9 (16.0) | 4.5 (1.7) | 3.1 (1.7) | 3.3 (1.7) | 87.0 (11.5) |
| p-value* | | <0.001 | 0.999 | 0.023 | 0.101 | 0.055 |

*One-way analysis of variance

**Student t-test.

DM: Decision-making preference subscale, URI: Upper respiratory tract illness clinical vignette, HBP: High blood pressure clinical vignette, MI: Myocardial infarction clinical vignette, IS: Information-seeking subscale

**Table 4. Associations between answers to the items of the additional vignette on preparedness to anticipate disease worsening and Autonomy Preference Index (API) subscale scores presented as median (quartile 1 –quartile 3).**

| Items from the additional clinical vignette | N (%) | API subscale scores | | | | |
|---|---|---|---|---|---|---|
| | | DM | URI | HBP | MI | IS |
| **1—In your opinion, who should take this decision in anticipation? (when the situation of sudden aggravation has not yet occurred)?** | | | | | | |
| · ... my own decision / ... after taking into consideration the doctor's advice | 29 (16) | 50 (42–63) | 5 (4–5) | 3 (2–4) | 4 (3–6) | 81 (75–91) |
| · ... my doctor and I decide together | 71 (38) | 50 (38–58) | 5 (4–5) | 3 (2–4) | 3 (2–4) | 91 (78–97) |
| · ... after taking my opinion into consideration / ... my doctor decide alone | 86 (46) | 38 (29–50) | 3 (2–5) | 2 (1–3) | 3 (2–3) | 88 (78–97) |
| p-value* | | <0.001 | 0.003 | <0.001 | <0.001 | 0.223 |
| **2—Is it important for you that your doctor discusses this decision with you in advance, in anticipation of a sudden worsening?** | | | | | | |
| · Yes, absolutely / Mostly yes | 177 (95) | 46 (33–58) | 4 (3–5) | 2 (2–4) | 3 (2–4) | 87 (78–97) |
| · Neutral / Mostly no / No, not at all | 9 (5) | 46 (38–50) | 5 (3–6) | 2 (0–3) | 3 (0–4) | 66 (59–84) |
| p-value** | | 0.707 | 0.390 | 0.780 | 0.463 | 0.010 |
| **3—Do you think it is possible to give your opinion on this decision when the situation has not yet arisen?** | | | | | | |
| · Yes, absolutely / Mostly yes | 130 (71) | 46 (33–58) | 4 (3–5) | 2 (2–4) | 3 (2–4) | 91 (78–97) |
| · Neutral / Mostly no / No, not at all | 53 (29) | 42 (33–52) | 4 (3–6) | 2 (1–3) | 3 (2–4) | 84 (75–97) |
| p-value** | | 0.079 | 0.282 | 0.121 | 0.845 | 0.250 |

* Kruskall-Wallis test

** Mann-Whitney's test.

DM: Decision Making, URI: Upper respiratory tract illness, HBP: High blood pressure, MI: Myocardial infarction, IS: Information Seeking.

with the API, as a comprehensive scale to guide doctor-patient communication in the context of advanced cancer.

Concerning the original 23-item API, a three-factor CFA model was found to have a better fit to the data than a two-factor model. While two factors were initially hypothesized [19], in a recently published study, the authors rigorously assessed the structural validity of the API adjusted for the setting of mental health and also found that a three-factor model provided a better fit for the data than a two-factor model [24]. This finding is also more consistent with the API scoring system, which distinguishes the vignette scores from the 6-item DM score, suggesting that these 15 items are likely to be linked to more than one factor. In agreement with previous findings, the desire for information factor was not or poorly correlated with the decision-making factors, [19,21,23] and it was the same items (4, 6 and 20; the reversely coded items) that were found to have low loadings [25]. No further analyses were performed to assess the fit of adapted models (*i.e.* without these reversely coded items, as in Bonfils et al [25]), as our aim was to adapt the classic version of API into French to facilitate comparisons between studies that have already used this version. However, our results are consistent with those of previous studies and suggest that it would be interesting to carefully reconstruct this instrument to enhance its psychometric properties.

Measurement invariance was assessed precisely to guarantee that group comparisons would be accurately interpreted. In this study, we did not find any measurement invariance issues related to age, sex, educational level or population studied. This means that, for example, the URI and HBP score differences observed between the two samples studied were not due to a different interpretation of one or several items in these two vignettes according to the setting.

They could result from a phenomenon of confusion, as there were many imbalanced characteristics between these two samples, or from real preference differences, but not from measurement error deriving from differential functioning of the API measurement tool between these two groups. In addition, thanks to the authors of the Australian study, [26] we were able to assess measurement invariance related to the language version (French and English) and found no issue for the 14 items, setting aside the vignettes (not included in the Australian study), meaning that a comparison of the IS and DM scores obtained using the two language versions can be accurately interpreted.

Finally, the assessment of the other psychometric properties of the original 23-item version of the API showed an acceptable level of internal consistency according to Cronbach's alpha coefficients, an acceptable level of test-retest reliability according to ICCs, good convergent validity and adequate *a priori* hypothesis testing for most of the API subscales.

Of course, this study is not without limitations. First of all, the size of the sample of patients with incurable cancer was too small to accurately assess the structural validity of the API. To circumvent the difficulty in recruiting patients with an incurable illness, we decided to recruit patients in a primary care setting and to assess measurement invariance of the API across settings. This study design enabled a sample size that guaranteed the accuracy of the assessment of the structural validity across the two settings. Another limitation concerned the fact that the three vignettes were not used in the Australian study and this precluded the assessment of the measurement invariance across language versions for these vignettes. In most of the studies on the API, these vignettes are not used, and to our knowledge, our study is the first where measurement invariance across language versions has been assessed for the IS and DM scores. Finally, it would have been interesting to assess measurement invariance according to other characteristics, like for example anxiety and depression which may influence the interpretation of some items of the API. However, due to time constraints we did not collect information on depression and anxiety level in the GP sample.

## 5. Conclusions

Our findings suggest that the French version of the API is valid and reliable in both general practice and oncology settings, and that accurate score comparisons can be made across age, sex, educational level, setting and English and French versions. The additional vignette developed provides interesting information on the patients' preparedness to anticipate disease deterioration, which can be of interest in the development of research on advance-care planning discussions to promote patient-centered care, ensuring that patient values guide all clinical decisions in the end-of-life period.

## Supporting information

**S1 Questionnaire. French version of the Autonomy Preference Index (API).**
(DOCX)

**S2 Questionnaire. Additional clinical vignette developed for use among patients with incurable cancer, to assess their preparedness to anticipate disease worsening.**
(DOCX)

**S1 Table. Frequency (%) of the answers to each item of the Autonomy Preference Index in both samples.**
(DOCX)

**S2 Table. Measurement invariance assessment for the three-factor model of the Autonomy Preference Index.**
(DOCX)

**S1 File. Dataset.**
(CSV)

**S2 File. Labels of variables in the dataset.**
(XLS)

## Acknowledgments

The authors would like to thank all the participants in the study, Lamisse Bouti (MSc) from the Research Center in Epidemiology and Population Health, INSERM U1018 and Paris-Sud University for her work in the data management and exploratory analyses processes, Lotfi Dahmane from the Public Health and Epidemiology Department of Paris-Sud University Hospitals for his help for data collection and regulatory procedures, Marie-Yvonne Guillard from the palliative care team for her help in the identification of patients eligible for the ONCO sample, Pr François Goldwasser head of the oncology department of the Cochin Hospital and all the staff members for their help in recruitment in the ONCO sample, and the general practitioners and Aude Benichou from the General Practice Department of the Paris-Sud University for their help in recruitment and data collection in the GP sample.

## Author Contributions

**Conceptualization:** Isabelle Colombet, Laurent Rigal, Miren Urtizberea, Pascale Vinant, Alexandra Rouquette.

**Data curation:** Laurent Rigal, Miren Urtizberea, Alexandra Rouquette.

**Formal analysis:** Alexandra Rouquette.

**Funding acquisition:** Isabelle Colombet.

**Investigation:** Isabelle Colombet, Laurent Rigal, Miren Urtizberea.

**Methodology:** Isabelle Colombet, Laurent Rigal, Miren Urtizberea, Alexandra Rouquette.

**Project administration:** Isabelle Colombet, Laurent Rigal, Pascale Vinant.

**Resources:** Isabelle Colombet, Laurent Rigal, Pascale Vinant.

**Software:** Alexandra Rouquette.

**Supervision:** Isabelle Colombet, Laurent Rigal, Pascale Vinant, Alexandra Rouquette.

**Validation:** Isabelle Colombet, Alexandra Rouquette.

**Writing – original draft:** Isabelle Colombet, Pascale Vinant, Alexandra Rouquette.

**Writing – review & editing:** Isabelle Colombet, Laurent Rigal, Miren Urtizberea, Pascale Vinant, Alexandra Rouquette.

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
