## [Decision Letter · Decision Letter 0]

17 Oct 2019

PONE-D-19-15483

Assessment of information preferences and desire to participate in decision-making: validity of the French version of the Autonomy Preference Index and its adaptation for patients with advanced cancer.

PLOS ONE

Dear Doctor Rouquette,

Thank you for submitting your manuscript to PLOS ONE. After careful consideration, we feel that it has merit but does not fully meet PLOS ONE’s publication criteria as it currently stands. Therefore, we invite you to submit a revised version of the manuscript that addresses the points raised during the review process.

Please, see the comments of two Reviewers appended at the bottom of this letter. At this point, Reviewer #1 recommends a minor revision, whereas Reviwer #2 recommends a major revision. After my own reading of the manuscript, I rather coincide with the points raised by Reviewer #2, who has provided extensive, and I think that useful feedback about the current version of your study. Because this might be considered as a major review, please notice that a resubmission will require an additional round of reviews, and that the final outcome of the process cannot be predicted at this point. If you decide to resubmit a revised version of your manuscript, please provide either a proper answer or rebuttal to each of the suggestions that were raised by the Reviewers.

We would appreciate receiving your revised manuscript by Dec 01 2019 11:59PM. To enhance the reproducibility of your results, we recommend that if applicable you deposit your laboratory protocols in protocols.io, where a protocol can be assigned its own identifier (DOI) such that it can be cited independently in the future. For instructions see: http://journals.plos.org/plosone/s/submission-guidelines#loc-laboratory-protocols

We look forward to receiving your revised manuscript.

Kind regards,

Angel Blanch, Ph.D.

Academic Editor

PLOS ONE

Journal Requirements:

Additional Editor Comments (if provided):

Reviewers' comments:

Reviewer's Responses to Questions

**Comments to the Author**

1. Is the manuscript technically sound, and do the data support the conclusions?

Reviewer #1: Yes

Reviewer #2: Partly

2. Has the statistical analysis been performed appropriately and rigorously? 

Reviewer #1: Yes

Reviewer #2: Yes

3. Have the authors made all data underlying the findings in their manuscript fully available?

Reviewer #1: Yes

Reviewer #2: No

4. Is the manuscript presented in an intelligible fashion and written in standard English?

Reviewer #1: Yes

Reviewer #2: Yes

5. Review Comments to the Author

Reviewer #1: Very interesting piece of research and rigorously presented. Below some comments:

Line 131: may you provide more information on medical data of the ONCO sample: years since diagnosis, type of treatments received (chemo, radio, surgery...); otherwise did you collect any measure of QoL, physical functioning and mood status? If not, I miss some discussion about how differences on these aspects may impact on preferences for DM, IS and preparedness to anticipate disease worsening... May this be a limitation regarding the generability of your findings?

Line 141: may you enhance description of how and by who patients from the ONCO sample were approached for data collection (what specialist consultation, known or unknown researcher to the patient...)?

Line 245: 8 'DM' items is maybe a typo, I think you mean IS

Line 291: I dont quite understand what you mean by 'disrupt coping mechanisms...', in turn I find this idea interestingly enough to develop it a bit further...

Line 292: 'educational initiatives' mean from the patient perspective, like psychoeducation, or from the healthcarer perspective, as in medical school programs?

Reviewer #2: The article Assessment of information preferences and desire to participate in decision-making: validity of the French version of the Autonomy Preference Index and its adaptation for patients with advanced cancer targets the validation of the Autonomy Preference Index (API). I highly appreciate such validation studies that apply confirmatory factor analysis and invariance testing. Although I see problems several major problems in the structure of the manuscript and the conclusion and handling of the results. In the next sections I further describe problems of the paper in detail.

Title: In my opinion the title is too long and should be shortened.

Abstract: I think the background should also include a quick description why such a validity study and enlargement is necessary. For me it is not clear what is meant by “a consensual French version”. In the results it should be indicated if the three-factors were correlated. As the last sentence in the results sections indicates that one new items was not correlated at all with the API, I believe the conclusion that additional items are of interest for the API is wrong, as this items clearly does not belong to the scale.

Regarding the open data statement: I don´t believe this is correct. Why are you not providing the data anonymized? Anonymized data can be shared and I would also like to so the scripts for analysis shared (e.g., in the OSF).

Introduction:

73ff you describe three questionnaires. You clearly choose API over the others for your validation study. I would appreciate a description why you choose this over the others.

In my opinion the introduction is too short and misses information about results from previous application of the API (e.g., The results of study [26] should be provided and discussed). This would help the reader to understand why you have chosen the API, why you aim a validity study and why you have developed additional items. Besides, I would like to read more about possible invariance in the introduction. After reading the introduction it is not clear, why you would (not) expect measurement invariance for gender, setting and age. A through explanation for that in the introduction (supported by previous results) is needed to understand the necessity and the aim of the study. I think the introduction needs a clear and literature based expansion on all the above named issues in order to be publishable. Also I highly recommend to at least provide research aims that clearly indicates what was assessed and expected (validation, reliability, invariance, etc.).

Methods:

L 102: ) is missing after HBP

As the term vignette can be understand differently it would be helpful if you present an example of the items you invented additionally.

I appreciate the process and description of translation.

2.3 Study samples describe the sample recruitment. What I miss in this section is a clrea description of the sample. How many participants were recruited in total? How is the age, gender, settings and educational level in this sample? Did the participants receive any incentive? How is further data distributed (health status etc.). This is all missing in the methods section and should be in detail included here.

L 159: what is qualitative data? Please elaborate further. I do not understand what of your data can/would be qualitative data (maybe due to confusing/inadequate description of the measurements: the measurement description is distributed all over the methods part and there is no general overview).

L 167: please state how many missing data you have and why you have missing data.

L 172ff: please indicate if the factors were correlated or not.

L 181: a short description of invariance would help reader that are not familiar with invariance to understand what the three invariance types are and how they are tested. I recommend a detail description here.

L 184: please rather indicate Mc Donalds omega (1999) as a reliability coefficient as it is a better estimate than Cronbachs alpha and can be directly estimated in the confirmatory models.

L 190ff: I do not understand how this (age, educational level, etc.) should be indicators for convergent and discriminant validity. Based on its definition a test for convergent validity requires a high correlation with a test that measures arguably the same construct while discriminant validity is indicated by a low correlation with a test that measures a different construct. Please re-consider if these are the correct indicators for testing convergent and discriminant validity – I don´t think so, but I am also no clinical expert – and if so please state why these indicators are correct and how they were testes (normally based on correlations, your test is not clearly understandable for me at this point). I also recommend quickly explaining convergent and discriminant validity.

Results:

Table 1 would rather be a good table for the methods section in order to describe the sample. I also don´t understand what the p-value in table 1 implicates. This clearly needs further explanation. I also do not really understand what the N is for API score, and the global judgments. Does the N displays the amount of people that have answered at all? That have answered in a specific way?

L 206: I would also highly appreciate to read more in the introduction if the different types of cancer might lead to differences in answering the questionnaire.

L 216: see my comment above. Why are there any missings – please describe?

In my opinion table 2 would be easier to understand as a graph (e.g. bar chart etc.) for each of the vignettes.

Figure 1. First, the picture quality is quite bad. Please indicate if all loadings were significantly. Was the correlation between decision making and information seeking (.10) really significant? As the introduction is quite vague and missing a lot of information for me it is not clear if the low/n.s. correlation between information seeking and the other scales is expected and how this was modeled before. Anyways from a psychometrical point of view this indicates that the three scales are not measuring the same thing or at least information seeking is something different than the other two scales. Hence I would highly recommend a careful reconstruction of the questionnaire. I recommend deleting items with very low loadings (e.g., below .30) and reconsider the third scale and its meaning for the API (as statistically it has nothing to do with the other scales). If that is expected or clinical meaningful in some way than it has to be made clear in the introduction.

L 231ff: I appreciate that it was tested for measurement invariance. Although due to the somehow confusing methods section it is not clear for me a) how big the single groups are and b) what cut-off were made (e.g., What cutoff was applied for age? And how was that cutoff selected). That has to be clearly stated.

L 244: as said above I highly recommend the use of McDonalds omega instead of alpha.

L 251: what is concurrent validity? This term is new here and was not introduced before. Why are you reporting a priori hypothesis down here, if you don´t report them in the end of the introduction? Please be consistent here.

Table 3. Due to the lacks in the introduction and methods I have problems of following what is what here. Where do I find the three scales of figure 1 in this table? Why is the clinical vignette now divided in three other scales even though it was one scale in figure 1. I highly recommend a better overview (as already stated) and consistent labeling. Besides, this table should be referenced before the invariance testing (maybe even as descriptive statistics in the methods) as I guess these are the groups you refer to in the invariance testing?

L 263ff: I don´t see any relationships in the table but rather quite un-informative descriptive statistics again. Please report correlations – I also recommend reporting them as heat maps to better visualize. Besides, as said above, please use consistent labeling.

L264ff: where is this relation you are talking about? How big are they? Please state r and p.

Discussion:

L277: For this conclusion I need to see a re-considered scale or a very good theoretical explanation why information seeking is not correlated with the other factors.

L279ff: you do not show that? How did you proved incremental validity of your vignette? Please state that differently or provide prove of any incremental validity over the existing scale.

L 285ff: I do not see the benefit of a vignette that is not correlated with nothing. This is no prove for incremental validity! It is not clear what this vignette is measuring in a psychometric way and I would highly recommend to re-consider its use at all.

L287ff: why did you thought that? Any empirical proof (please cite it here) or just a feeling?

L296ff: why? Please elaborate on how this vignette can do that. From a psychometrical point of view I rather see it questionable (due to its correlational results) and doubt its meaning.

L310ff: I appreciate that you mention loading problems and the low correlations between the subscales. But I think this really implies further analysis. The reason not to dig too deep into modelling issue because of “adapting the classic version” seems quite weak. Arguably the classic version comes along with problems (or at least some items have major problems). So why not improve the scale in order to measure the construct even better, more valide and more reliable?

6. PLOS authors have the option to publish the peer review history of their article (what does this mean?). If published, this will include your full peer review and any attached files.

Reviewer #1: Yes: Alejandra Cano Carmona

Reviewer #2: No

---

## [Author Response · Author response to Decision Letter 0]

21 Nov 2019

A rebuttal letter that responds to each point raised by the academic editor and reviewers has been uploaded as separate file and labeled 'Response to Reviewers'.

---

## [Decision Letter · Decision Letter 1]

20 Dec 2019

PONE-D-19-15483R1

Validity of the French version of the Autonomy Preference Index and its adaptation for patients with advanced cancer.

PLOS ONE

Dear Doctor Rouquette,

Thank you for submitting your manuscript to PLOS ONE. After careful consideration, we feel that it has merit but does not fully meet PLOS ONE’s publication criteria as it currently stands. Therefore, we invite you to submit a revised version of the manuscript that addresses the points raised during the review process.

We would appreciate receiving your revised manuscript by Feb 03 2020 11:59PM. To enhance the reproducibility of your results, we recommend that if applicable you deposit your laboratory protocols in protocols.io, where a protocol can be assigned its own identifier (DOI) such that it can be cited independently in the future. For instructions see: http://journals.plos.org/plosone/s/submission-guidelines#loc-laboratory-protocols

We look forward to receiving your revised manuscript.

Kind regards,

Angel Blanch, Ph.D.

Academic Editor

PLOS ONE

Reviewers' comments:

Reviewer's Responses to Questions

**Comments to the Author**

1. If the authors have adequately addressed your comments raised in a previous round of review and you feel that this manuscript is now acceptable for publication, you may indicate that here to bypass the “Comments to the Author” section, enter your conflict of interest statement in the “Confidential to Editor” section, and submit your "Accept" recommendation.

Reviewer #2: (No Response)

2. Is the manuscript technically sound, and do the data support the conclusions?

Reviewer #2: Yes

3. Has the statistical analysis been performed appropriately and rigorously? 

Reviewer #2: Yes

4. Have the authors made all data underlying the findings in their manuscript fully available?

Reviewer #2: Yes

5. Is the manuscript presented in an intelligible fashion and written in standard English?

Reviewer #2: Yes

6. Review Comments to the Author

Reviewer #2: Thank you for re-working your paper. I think the paper is much better now, easier to read and shows interesting, valuable and good results. I thank the authors for working on all my prior suggestions.

I have two minor points that has to be addressed prior to publication as I think the authors made a mistake here:

Added information (line 33-34): “A three-factor confirmatory factor model (two correlated factors related to decision-making and an uncorrelated factor related to information-seeking preferences” This is not what you report as final model in figure 1 (these are three correlated factors). Is this the final model you report? I either suggest re-working figure 1 towards your final model (as this is the only model you display it is misleading) if it is not the final model. If figure 1 is the final model you have to re-write the passage accordingly and report three correlated factors. For clarification: the factor information seeking shows only small/no correlation with the other factors. Nevertheless, the model has to be declared as three-correlated factors as you allow the factors to correlate (maybe this was the mistake here?). This indicates: three-correlated factor model that as a result shows that one factor is not/smally correlated with the others.

Added information in Line 243-244: Again the factors were allowed to be correlated but as a result they only showed small/no significant correlation. Please be consistent here.

7. PLOS authors have the option to publish the peer review history of their article (what does this mean?). If published, this will include your full peer review and any attached files.

Reviewer #2: No

---

## [Author Response · Author response to Decision Letter 1]

21 Dec 2019

Reviewer #2: Thank you for re-working your paper. I think the paper is much better now, easier to read and shows interesting, valuable and good results. I thank the authors for working on all my prior suggestions.

I have two minor points that has to be addressed prior to publication as I think the authors made a mistake here:

Added information (line 33-34): “A three-factor confirmatory factor model (two correlated factors related to decision-making and an uncorrelated factor related to information-seeking preferences” This is not what you report as final model in figure 1 (these are three correlated factors). Is this the final model you report? I either suggest re-working figure 1 towards your final model (as this is the only model you display it is misleading) if it is not the final model. If figure 1 is the final model you have to re-write the passage accordingly and report three correlated factors. For clarification: the factor information seeking shows only small/no correlation with the other factors. Nevertheless, the model has to be declared as three-correlated factors as you allow the factors to correlate (maybe this was the mistake here?). This indicates: three-correlated factor model that as a result shows that one factor is not/smally correlated with the others.

Yes you are right, we fixed this mistake in the abstract as the model depicted in the figure 1 is the final model (line 31-34) : “A three correlated factors confirmatory model (two factors related to decision-making and a factor related to information-seeking preferences) showed an acceptable fit on the whole sample and no measurement invariance issue was found across settings, age, sex and educational level.”

Added information in Line 243-244: Again the factors were allowed to be correlated but as a result they only showed small/no significant correlation. Please be consistent here.

 � We also modified this sentence to be more consistent (line 357-359) : “In agreement with previous findings, the desire for information factor was not or poorly correlated with the decision-making factors, [19,21,23] and it was the same items (4, 6 and 20; the reversely coded items) that were found to have low loadings [25].”

 � We used PACE to ensure that the Figure 1 (APIvalid_Figure1.tif) meet PLOS requirements.

---

## [Editor Report · Decision Letter 2]

31 Dec 2019

Validity of the French version of the Autonomy Preference Index and its adaptation for patients with advanced cancer.

PONE-D-19-15483R2

Dear Dr. Rouquette,

We are pleased to inform you that your manuscript has been judged scientifically suitable for publication and will be formally accepted for publication once it complies with all outstanding technical requirements.

With kind regards,

Angel Blanch, Ph.D.

Academic Editor

PLOS ONE
---

## [Editor Report · Acceptance letter]

3 Jan 2020

PONE-D-19-15483R2 

Validity of the French version of the Autonomy Preference Index and its adaptation for patients with advanced cancer. 

Dear Dr. Rouquette:

I am pleased to inform you that your manuscript has been deemed suitable for publication in PLOS ONE. Congratulations! Your manuscript is now with our production department. 

With kind regards,

on behalf of

Dr. Angel Blanch 

Academic Editor

PLOS ONE